# New Bioinformatic Insight into CD44: Classification of Human Variants and Structural Analysis of CD44 Targeting

**DOI:** 10.3390/ijms26209886

**Published:** 2025-10-11

**Authors:** Wiktoria A. Gerlicz, Aleksandra Olczak, Aneta M. Białkowska, Aleksandra Twarda-Clapa

**Affiliations:** Institute of Molecular and Industrial Biotechnology, Faculty of Biotechnology and Food Sciences, Lodz University of Technology, Stefanowskiego 2/22, 90-537 Lodz, Poland

**Keywords:** CD44, hyaluronic acid, HA, osteopontin, OPN, Link domain, alternative splicing, small-molecule inhibitors, biomarkers, cancer stem cells

## Abstract

The cluster of differentiation 44 (CD44) is a member of the hyaluronic acid (HA) receptor family of cell adhesion molecules. Besides HA, this transmembrane protein also serves as a receptor for other components of the extracellular matrix (ECM), including fibronectin, collagen, and osteopontin (OPN). The CD44-HA axis is involved in a wide range of physiological and cancer-related processes, particularly in cell adhesion and migration, lymphocyte activation, as well as tumour progression and metastasis. The possibility of modulating the CD44-HA interaction with a pharmacological inhibitor has therefore been recognized as an emerging anti-cancer strategy. With its expression in a wide variety, CD44 has also become the most common surface biomarker of cancer stem cells. Due to the rapid progress of research on this crucial receptor, some published and deposited variants were often poorly described or lacked accession numbers in the available protein databases, which created confusion and hindered relevant research. In this work, we attempted to examine the protein sequences of the known CD44 variants and match them between the two UniProt and the National Centre for Biotechnology Information (NCBI) Protein databases. The deposited sequences were aligned to the CD44 canonical sequence and grouped based on the observed differences. Analysis of CD44–ligand experimental structures available in the Protein Data Bank (PDB) was also performed to identify the most promising small-molecule inhibitors of the CD44-HA interaction.

## 1. Introduction

The term cluster of differentiation 44 (in short, CD44) describes a family of type I transmembrane glycoproteins, which are expressed on the cell surface in a variety of tissues. The molecule was first described in 1980 as a “human brain leucocyte antigen” by Dalchau et al. with the mAb F10-44-2 [1,2] and further characterized in 1983 by Haynes et al. [3], and in 1987 by Borche et al. [2]. During the Third International Workshop and Conference on Human Leukocyte Differentiation Antigens, a selection of broad-specificity antibodies, including F10-44-2, was screened on various haematopoietic cell lines. Based on the analyses performed, a new cluster of an unknown function was distinguished and assigned the name “CDw44” [4]. In the following years, several molecules such as Inlu-related p80 glycoprotein [3,5,6], homing cell adhesion molecule (H-CAM) [5], ECMR-III (extracellular matrix receptor type III) [5,6], phagocytic glycoprotein-1 (Pgp-1) [5,6,7], Hermes-1 antigen [6,8], gp90^Hermes^ [9], leucocyte homing receptor [10] were found to be synonymous with CD44. Around the same time, the antigen was also recognized as a member of the cartilage link protein family [11].

### 1.1. CD44-HA Signalling

The current CD44 research primarily focuses on its interactions with ligands. The best-studied processes are those initiated by hyaluronan (HA) binding. HA is an unsulfurated glycosaminoglycan (GAG) consisting of alternating 1,4-β-linked d-glucuronic acid and 1,3-α-linked N-acetyl-d-glucosamine monomers [12]. The polymer occurs naturally throughout the human body as one of the main components of the extracellular matrix (ECM), and is cleared from the bloodstream, e.g., by the hyaluronic acid receptor for endocytosis (HARE, or stabilin-2/Stab2). The polysaccharide interacts with the activated CD44 receptor to modulate various cellular processes. CD44 shows three states of activation—inactive, inducible active, and constitutively active—related to the receptor’s capacity to bind hyaluronic acid [13]. The degree of CD44 activation is suggested to be tightly linked to the number of HA-CD44 interactions in a cell. The activated CD44 is predominantly expressed in cancerous tissues, leading to tumour cell survival, increased proliferation, enhanced motility, and cytoskeletal changes. In normal cells, the ligand binding affinity is much lower but can be induced through interactions with cytokines and inflammatory agents [14]. The exact mechanism of CD44 activation is unknown; however, it is assumed to be regulated through post-translational modifications (PTMs), CD44 clustering, and cytoskeletal associations [13].

Another factor necessary for the HA-CD44 binding and signalling is a functional intracellular domain (ICD). The cytoplasmic tail consists of 72 amino acid residues, constituting four ligand-binding structures: the 4.1/ezrin/radixin/moesin (FERM) domain that binds the ezrin/radixin/moesin (ERM) proteins, the ankyrin-binding domain, the PSD-95/Dlg/ZO-1 (PDZ) binding domain, and a basolateral targeting association site [15]. Through interactions with cytoskeletal proteins, these structures are responsible for downstream cellular signalling. Without this highly conserved region, interactions between the receptor and hyaluronan are impaired or, in some cases, impossible [15]. Upon CD44-HA binding, the ICD binds and activates cytoskeletal proteins (ERM, ankyrin, and PDZ), initiating a cascade of signalling reactions modulating one of four types of pathways— Mitogen-activated protein kinase (MAPK), Phosphoinositide 3-kinase (PI3K)/Protein kinase B (Akt), and ROCK-GTPase-activating-like protein 1 (IQGAP1)—and filamentous actin rearrangement (Figure 1). CD44 does not show intrinsic kinase activity; instead, it promotes phosphorylation of the bound intracellular proteins by protein kinases [16].

Activation of FERM-bound ERM proteins initiates a signalling cascade along the Raf kinase (Raf)/Ras GTPase (Ras)/MAPK/Extracellular signal-regulated kinase (ERK) pathway. As a result, cell proliferation and migration are induced, but in some cases, apoptosis can also be increased [17]. Another protein that can be recruited upon HA binding is ankyrin. Ankyrin binds to the inositol 1,4,5-triphosphate (IP3) receptor (IP3R) and regulates receptor localization, as well as its binding affinity towards IP3 and calcium cations [18]. The interaction between CD44, ankyrin, and IP3R promotes Ca^2+^ mobilization and release from intracellular stores [19]. The free calcium ions bind to calmodulin (CaM) to create a calcium–CaM complex, which in turn activates CaM-regulated kinase II (CAMKK2) [20].

Apart from being responsible for downstream signalling, CAMKK2 can also phosphorylate cytoskeletal filamins to stimulate cell migration [16]. The ICD-bound ankyrin can also activate PI3K. PI3K facilitates the conversion of phosphatidylinositol 4,5-bisphosphate (PIP2) to phosphatidylinositol 3,4,5-trisphosphate (PIP3), followed by the phosphorylation of small GTPases belonging to the Rho family, mainly Rho, Rac, and cell division cycle 42 (Cdc42). The active Rho GTPases interact with their downstream targets (mammalian Diaphanous-related formin (mDia), Wiskott–Aldrich syndrome protein-family verprolin homologous protein (WAVE), p21-activated kinase (PAK), and Rho-associated coiled-coil kinase (ROCK)) to increase cytoskeletal repatterning. Nucleation of filamentous actin by mDia, as well as the positive interactions of WAVE with actin-related protein 2/3 complex (Arp2/3) and ROCK with myosin II light chain (MLC), lead to increased polymerisation and contraction [21]. PAK and ROCK also regulate the filamin turnover by suppressing cofilin (via phosphorylation of LIM-motif containing kinase, LIMK) or (for ROCK only) myosin light chain phosphatase (MLCP) inhibition [21]. ROCK can also be a trigger for increased cell survival. The active kinase stimulates Gab-1, which in turn activates the PI3K/Akt regulatory pathway [16]. Additionally, a similar effect is also achieved through the IQGAP1 signalling pathway. IQGAP1 forms a complex with ERK2 and upregulates gene transcription [16]. As previously mentioned, all the mentioned interactions, as well as other unmentioned ones, are dysregulated in tumour cells, leading to enhanced carcinogenesis and resistance to therapy.

**Figure 1 ijms-26-09886-f001:**
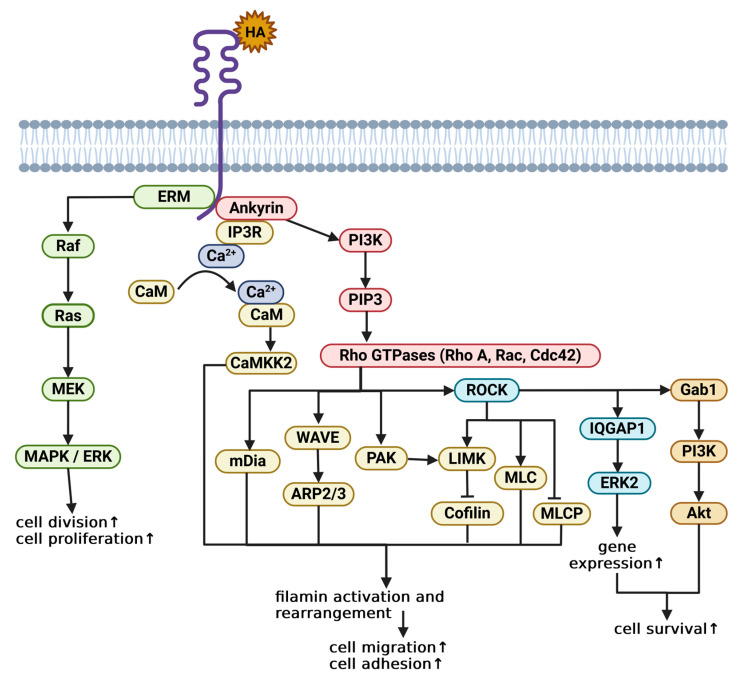
Overview of crucial metabolic pathways regulated via CD44-HA interactions. Green—MAPK/ERK pathway; yellow—filamentous actin polymerization; blue—IQGAP1 pathway; orange—PI3K/Akt pathway. Stimulatory effect on cell processes is indicated with an upward arrow (↑). Based on [16,20,21]. Created with BioRender.com.

### 1.2. The Role of CD44 and Osteopontin Interactions in Biomineralization

Except for HA, osteopontin (OPN) is another canonical extracellular ligand for CD44 [22]. OPN is an aspartic acid-rich, approximately 300-residue multi-domain glycoprotein that can be highly phosphorylated, imparting an acidic character. It belongs to the family of non-collagenous proteins known as SIBLING (small integrin-binding ligand, N-linked glycoprotein). In humans, OPN is encoded by the *Spp1* gene located on the long arm of chromosome 4, region 22 (4q1322.1) [23]. Osteopontin contains several adhesive domains that enable it to interact with cell surface receptors (CD44 and integrins) and ECM components (collagen, fibronectin, and osteocalcin) (Appendix A). The best-known domains enabling interactions with receptors include the domain containing the RGD sequence (arginine–glycine–aspartate), which binds to integrins; the SVVYGLR domain (serine–valine–valine–tyrosine–glutamic acid–leucine–arginine), which binds to the α9β1 integrin after exposure through thrombin cleavage; and the LPV domain (leucine-proline-valine) that binds to the α4β1 integrin. Additionally, OPN is associated with matrix metalloproteinases 3 and 7 (MMP-3 and MMP-7). Concerning the OPN termini, the C-terminal binds two heparin molecules and CD44 variants, while the N-terminal includes integrin receptor binding zones (Appendix A).

The expression of OPN is regulated by a wide array of cytokines, hormones, and growth factors, which influence gene transcription, translation, and post-translational modifications (e.g., transcription factors—Runx2 and Osterix; inorganic phosphate; hormones—glucocorticoids; vitamins—retinoic acid; and inflammatory mediators—TNFα, IL-1β, and TGFβ) [24]. Additionally, OPN expression increases in response to mechanical stress [25,26].

OPN is involved in various pathological and physiological events, including bone remodelling, biomineralization, wound healing, apoptosis, and tumour metastasis [27]. The role of the interaction with the CD44 receptor is significant for each of these functions. Concerning biomineralization, it has been demonstrated that OPN is localized in the mineralized collagen matrix, particularly in cement lines and lamina limitans, which are structures formed through the adhesive interactions of bone cells during bone formation and remodelling. OPN facilitates the adhesion of osteoclasts and osteoblasts to the bone matrix and stimulates osteoclasts, through interaction with integrin αvβ3, to migrate, increase podosome formation, and resorb bone [27].

CD44/OPN interaction depends on the splicing variant of CD44 [28]. One study demonstrated that CD44v3–v6, a marker associated with cancer cells, facilitates the adhesion and migration of osteopontin [22]. However, other studies reported no interaction between the standard form and some common CD44 isoforms, such as CD44H, CD44E, CD44v3, CD44v3–v6, and osteopontin [29,30]. As reported by Katagiri et al., and later by Lee and colleagues, only isoforms containing V6 and/or V7 exons were able to bind OPN [30,31]. The role of each exon and the exact binding mechanism remains unknown. Chellaiah and Hruska (2002) demonstrate that the migratory effects of osteoclasts are mediated through both integrin α_v_β_3_ and CD44, with a cooperative interaction between these two receptors (Figure 2) [32]. Similar conclusions were presented in the study of Chellaiah et al. (2003) [33]. Osteoclast-like cells derived from the peripheral blood of osteopetrotic patients and healthy individuals exhibited similar morphology but showed alterations in CD44 expression. These osteoclasts were defective in bone resorption function [34]. Spatial and temporal patterns of OPN/CD44 expression were observed in healing fractures of rat femora. In the remodelling callus, CD44 expression was detected on the basolateral plasma membrane of osteoclasts and osteocyte lacunae, but not in cuboidal osteoblasts. It has been demonstrated that OPN is the primary ligand for CD44 on bone cells during the remodelling phase of healing fractures, and the CD44/OPN interaction has clinical implications for the repair of skeletal tissues [35]. The decrease in CD44 expression in osteopetrotic patients [34] and in OPN^−/−^ mice, which are mildly osteopetrotic, suggests a role of CD44 in osteoclast function and bone modelling. Chellaiah et al. (2003) [33] elucidate the essential roles of OPN in osteoclast function, providing new insights into the roles of α_v_β_3_ and CD44 in osteoclast function and disordered bone modelling. They demonstrated that the OPN/α_v_β_3_-generated outside-in Rho signalling pathway is required for the surface expression of CD44, as well as the formation of the CD44-associated signalling complex. Osteoclast motility necessitates both α_v_β_3_ and CD44 receptors. The surface expression of CD44 can modulate multiple signalling pathways essential for osteoclast motility. The addition of external OPN partially restores the resorptive function of osteoclasts, highlighting the significance of autocrine OPN in osteoclast activity [33]. However, externally added OPN cannot access the OPN secreted by osteoclasts located in resorption lacunae. The intracellular form of OPN (iOPN), a crucial component of the CD44-ERM complex, plays a key role in the migration of osteoclasts [27,36].

Knockdown of CD44 leads to reduced mineralization, as demonstrated by Baugh et al. (2019) [36]. They demonstrated that reducing CD44 expression using siRNA leads to decreased calcification of valve interstitial cells (VICs) in an in vitro model. The authors observed that the knockdown of CD44 results in increased expression of OPN, suggesting that CD44 may play a role in regulating mineralization through interaction with OPN. Specifically, reducing CD44 expression resulted in smaller mineral nodules on polyacrylamide (PAAM) gels with HA, but did not affect the size of nodules on collagen gels. The study utilized an in vitro model based on PAAM gels, which allow precise control over stiffness and ECM binding sites. VICs seeded on PAAM gels of varying stiffness (5, 20, and 35 kPa) grew calcified nodules over the 3-week experiment, regardless of the binding protein. However, the knockdown of CD44 led to smaller nodules on HA gels, suggesting that ECM-to-cell signalling through CD44 may play a role in exacerbating valve tissue calcification. Moreover, RNA expression analysis in VICs with CD44 knockdown revealed differences in the expression of VIC activation markers (TGF-β1 and SMA) and the osteogenic marker OPN, providing insight into the role of CD44 in the mineralization pathway. In VICs with CD44 knockdown, an initial increase in OPN expression was observed compared to the control, suggesting that CD44 is involved in the early stages of mineralization. This study suggested that CD44, through its interaction with OPN, may play a crucial role in the biomineralization process and could be a potential therapeutic target for treating calcification-related diseases such as aortic valve disease. Finally, Yang et al. demonstrated that another player in atherosclerotic calcification, bone morphogenetic protein 2 (BMP-2), is regulated by CD44 by showing that BMP-2 could enhance migration and proliferation of hypoxia-induced vascular smooth muscle cells (VSMCs) via the actin/CD44/matrix metalloprotease-2 (MMP-2) molecular pathway [37]. The involvement of CD44 in the OPN/MMPs pathways should be further explored in other pathophysiological states related to calcification, including medial arterial calcification (MAC) [38].

Recently, it has been demonstrated that CD44 displayed a compensatory effect in the absence of immunoglobulin superfamily 11 (IgSF11), a calcium-dependent cell adhesion molecule. Mice with IgSF11 knockout have recently been shown to exhibit high bone mass. The ability of CD44 to compensate for IgSF11 deficiency was apparently due to the interaction of both CD44 and IgSF11 with PSD-95, a scaffolding protein on the cytoplasmic face of cell adhesions [39]. This data suggests that, in osteoclasts, CD44, with its ability to interact with extracellular OPN, might also interact with cell adhesion molecules such as IgSF11 in the coordination of matrix binding and cell fusion. In the case of osteoclasts, where cell fusion (and fission) varies depending on the substrate, this may be a crucial regulatory activity [40,41]; If so, this regulation involves redundancies. Further studies will be required to examine and clarify these questions.

Other studies have recently confirmed that structural constraints affect the interaction between OPN and CD44. Surface plasmon resonance (SPR) experiments revealed that OPN requires immobilization or the addition of heparin for strong ligation to CD44. Heparin binding prompts the unfolding of a core element in OPN, which is essential for efficient CD44 interaction. This conformational adjustment is primarily governed by electrostatic forces between heparin and charged patches along the protein backbone, thereby balancing the entropic losses incurred through ligand engagement [42].

The integrin α_9_β_1_ competes with the OPN-CD44 engagement, while integrin α_V_β_3_ reflects additive binding, suggesting that the CD44 contact sites on OPN are downstream of the RGD motif but overlap with the SVVYGLR domain. Hyaluronate, which binds to the far N-terminal domain of CD44, does not affect the OPN-CD44 interaction, indicating that these two ligands bind to distinct receptor domains. Further experiments using enzyme-linked immunosorbent assay (ELISA) and pull-down assays corroborated the SPR results. The addition of low concentrations of heparin to OPN enhanced its binding to CD44, while high concentrations of heparin inhibited the interaction. This suggests that heparin provides the required structural constraint for OPN, facilitating its interaction with CD44. The presence of divalent cations, such as manganese, also contributes to the interaction, although it is not required. The structural flexibility of OPN, resulting from its largely unstructured conformation, enables it to associate with diverse binding partners rapidly. Although OPN does not fold into a single defined structure, it comprises distinct local secondary structure elements with reduced conformational flexibility. These compacted domains encompass the binding sites for integrin αVβ3 and heparin [43,44]. The interaction between OPN and CD44 is contingent on accessibility to these compacted core regions, which may be achieved via heparin binding. In Summary, the interaction between CD44 and osteopontin is facilitated by structural constraints that enhance binding efficiency. Heparin plays a crucial role in this process by inducing conformational changes in OPN that expose binding sites for CD44. Understanding these interactions provides insights into the molecular mechanisms underlying tissue remodelling, immune responses, and cancer progression, and may inform therapeutic strategies targeting the OPN-CD44 interaction [42].

**Figure 2 ijms-26-09886-f002:**
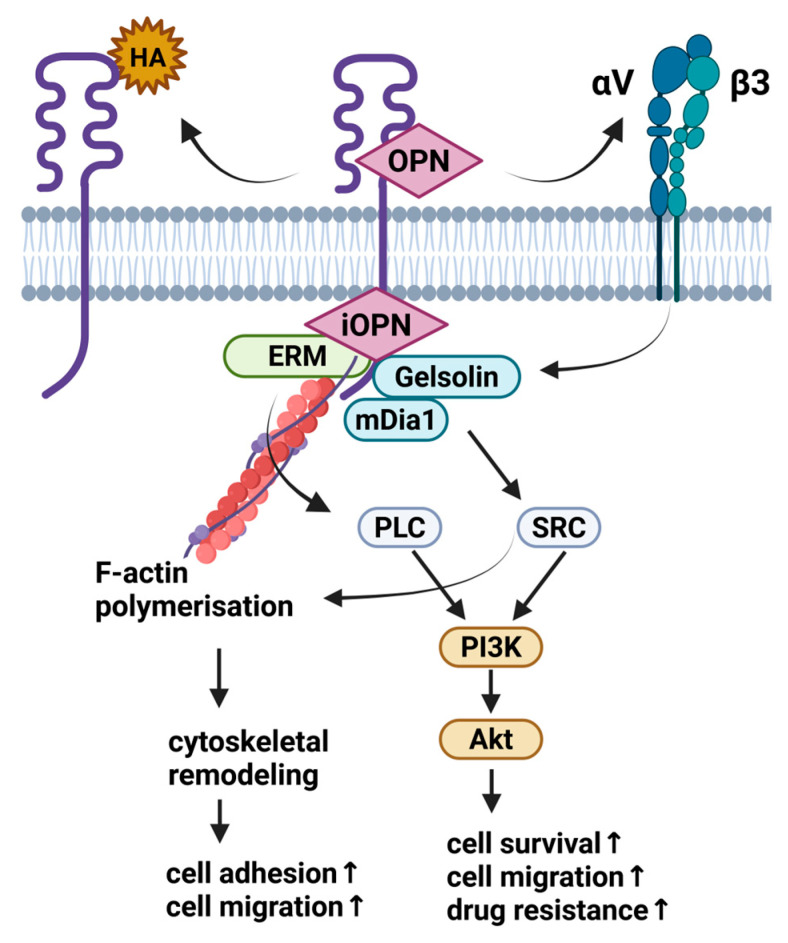
Overview of the metabolic pathways regulated via CD44-OPN interactions. Violet—OPN/iOPN; blue/marine—integrins aV and b3, and mDia1/Gelsolin; lilac—PLC/SRC; green—ERM; orange—PI3K/Akt pathway. Stimulatory effect on cell processes is indicated with an upward arrow (↑). Based on [16,45]. Created with BioRender.com.

### 1.3. Overall CD44 Structure

The CD44 gene is encoded on human chromosome locus 11p13 [46]. As initially proposed by Screaton et al., the coding sequence consists of 19 exons [47] (Appendix A). A CD44 receptor is made up of an extracellular domain ((ECD), also known as the ectodomain) (exons 1–16), a transmembrane domain (TMD) (exon 17), and an intracellular domain (exon 18 or 19) (Figure 3). The ECD can be further divided into three main parts—a highly conserved common region (exons 1–5) stabilized by three disulfide bonds, as well as a variable region (exons 6–14, also denoted as V2–V10), and a stalk region (exons 15–16) (Figure 3). In mice, an additional variable exon, V1, is located before V2 [16]; therefore, the numbering of the exons is slightly different. The variable exons (6–14) undergo alternative splicing and, therefore, vary depending on the CD44 variant [48]. The standard isoform (NP_001001391.1; P16070-12) contains no variable region, while the canonical (NP_000601.3; P16070-1) contains all nine variable exons, making it the longest isoform.

Two versions of the cytoplasmic domain (long-tail or short-tail) have also been reported. The majority of isoforms contain exon 19 encoded by 67 amino acid residues followed by the 3′ untranslated region (UTR), as first reported by Goldstein et al. [9]. Similarly to the long-tail version, the cytoplasmic domain begins with the last three amino acids of exon 17 (RRR for long tail and RRS for short tail). The CD44st isoforms are commonly differentiated by the inclusion of exon 18 (present only in the short-tail variants), which encodes an alternative stop codon. However, there is no coding sequence specific to exon 18; instead, the 3′UTR was originally indicated as the differentiating element between the two ICD versions [47]. A recent paper by Skandalis reports a slightly longer short-tail sequence with two additional amino acids [15]. Interestingly, both Skandalis and Screaton cite the same reference: Goldstein et al., 1989 [9]. Unfortunately, the additional amino acid residues were not specified by Skandalis, nor were we able to deduce them from the cited sequences (AAA36138.1; M25078.1).

## 2. Results and Discussion

### 2.1. Recognized CD44 Isoforms and Their Nomenclature

The nomenclature regarding the human CD44 variants is not standardized; therefore, the names and abbreviations vary depending on the protein database and scientific publication. This problem and its consequences for research development have already been reported [49,50]. In this section, we aim to present the update of CD44 variant classification and decipher the complexity of the current nomenclature.

First, recognized and numbered CD44 variants exist in the databases. UniProt database entry P16070 lists 19 such isoforms (denoted as P16070-1 to P16070-19), as well as 19 computationally mapped potential-isoform sequences, and the National Center for Biotechnology Information (NCBI) Protein database [50] lists, as of 19 May 2025, 38 recognized variants (numbered 1–38) and 5 predicted structures (denoted as X2, X6, X7, X11, and X22). Before the reclassification, NCBI listed eight biological isoforms (NCBI 1-8) and twenty-seven computational ones (X1–X27). Currently, 23 predicted structures have been made official variants, and 7 entirely new ones have been introduced. All new isoforms (NCBI 9-38) are listed as derived from two chromosome 11 partial sequences (AL356215.11 and AL356215.14) submitted in 2012 by a member of the Human Genome Consortium. The reason for such a significant change being made now, based on a 13-year-old submission, is unknown to us. Nevertheless, the fact remains that the number of recognized isoforms, as well as their designated abbreviations, varies between the UniProt and NCBI databases, which can create confusion and hinder relevant research.

As previously stated, we attempted to examine the biodiversity of the known CD44 variants and match them between the two databases. Each sequence was aligned to the canonical sequence, and grouped based on the observed differences—first based on the exons it contained, and second by any additional differences from the full-length. A selection of the obtained data (excluding the computationally proposed isoforms, partials, and the ungrouped NCBI isoforms) is shown in Table 1. An extended version of Table 1, which includes all non-partial isoforms currently available in NCBI and UniProt databases, can be found in the Appendix A (Appendix A for complete sequences; Appendix A for partials; and Appendix A for sequencing methods). The differences between the variants have been described following the Human Genome Variation Society (HGVS) nomenclature recommendations [51], using one code amino acid abbreviations. A brief explanation of this system can be found in Table 2.

**Table 1 ijms-26-09886-t001:** A selection of non-partial CD44 isoforms from NCBI and UniProt databases grouped by the exons they contain, with additional differences from the canonical form (P16070-1; NP_000603.1), and the oldest reference indicated for each structure (an extended version available in Appendix A). For some UniProt entries, the first reference was an archived Ensembl page or the now-defunct International Protein Index (IPI) database. In such cases, the reference has been noted as a question mark: (?).

Exon(s) Spliced Out	Abbreviation(s)	[aa]	Additional Differences	NCBI Accession	UniProt Accession	Ref.
**N/A** (Full-length form)	CD44v2-10 canonical NCBI 1 UniProt 1	742		NP_000601.3	P16070-1	[47,52]
	unnamed isoform	742	p.K417R p.I479T	AAB13628.1 KAI4070736.1		[47,52]
	unnamed isoform	742	p.S109Y p.T241A p.K417R p.I479T p.D494N	CAB61878.1		[53]
	CRA_d	742	p.K417R	EAW68148.1 EAW68147.1 KAI2559389.1		[54]
	NCBI 9	743	p.345_346insA	NP_001427253.1		[55]
	NCBI 10	741	p.A428del	NP_001427254.1		[56]
**Exons 2–19** p.23_742delinsGVGRRKS	CD44SP UniProt 2	29		AAB27917.1	P16070-2	[57]
**Exons 3–19** p.78_742delinsSST	unnamed isoform	80		AAH52287.1 KAI2559398.1 KAI4070737.1		[54]
**Exons 3–19** p.78_742delinsSLHCSQQ SKKVWAEEKASDQQW QWSCGGQKCGGQKAK WTQRRGQQVSGNGAF GEQGVVRNSRPVYDS	CD44sol (soluble) CD44RC, CRA_g NCBI 5 UniProt 19	139		NP_001001392.1 AAC70782.1 EAW68152.1 KAI2559396.1 KAI4070741.1	P16070-19	[58]
**Part of exon 5** p.192_223delinsA	UniProt 3	711	-		P16070-3	(?)
	NCBI 11	712	p.345_346insA	NP_001427255.1		[55]
**Part of exon 5** p.192_223delinsA **Exon V6** p.385_428delinsT	unnamed isoform	668	p.I479T	AAB13626.1		[47,52]
	CRA_h UniProt 16	668		EAW68153.1	P16070-16	[59]
**Exon V2** p.223_266delinsS	CD44v3-10 epidermal epican NCBI 2 UniProt 4	699		NP_001001389.1	P16070-4	[60]
	epican	699	p.E410V p.I479T	CAA47271.1		[60]
	CRA_f	699	p.K417R	EAW68150.1 KAI2559390.1		[54]
	unnamed isoform	699	K417R p.I479T	AAH04372.1 KAI4070734.1		[54]
	NCBI 12	700	p.345_346insA	NP_001427256.1		[55]
	NCBI 13	699	p.345_346insA p.A428del	NP_0001427257.1		[55]
	NCBI 14	696	p.A123_E126del p.345_346insA	NP_001427258.1		[55]
**Exon V2** p.223_266delinsS **Exons V7–19** p.428_742delinsGDCGS MAWVKKYFSFIFL	NCBI 36	403	p.345_346insA	NP_001427280.1	-	[55]
	NCBI 37	402		NP_001427281.1	-	[55]
**Exons V2–V3** p.223_308delinsI	NCBI 20	658	p.345_346insA	NP_001427264.1		[55]
	NCBI 21	657		NP_001427265.1		[55]
**Exons V2–V4** p.223_385delinsI	NCBI 26	580		NP_001427270.1		[55]
	NCBI 27	579	p.A428del	NP_001427271.1		[55]
**Exon V2** p.223_266delinsS **Exons V4–V7** p.308_472delinsN	NCBI 31 CRA_j	535		NP_001427275.1 EAW68155.1		[59]
**Exons V2-V7** p.223_472delinsN	CD44v8-10 epithelial keratinocyte CD44E CRA_e NCBI 3 UniProt 10	493		**NP_001001390.1** EAW68151.1 EAW68149.1 KAI2559391.1	P16070-10	[61,62]
	CD44R1	493	p.I479T	CAA40133.1 AAB13627.1 KAI4070739.1	-	[47,52]
**Exons V2–V7** p.223_472delinsN **Exons V9–V10** p.506_604delinsR	UniProt 14	396		-	P16070-14	[57]
	CRA_i	396		EAW68154.1	-	[59]
	CD44R5	395	p.I479T	AAB27919.1	-	[57]
**Exons V2–V7** p.223_472delinsN **Exon V10** p.536-604delinsR	CD44R4 UniProt 13	425	p.I479T	AAB27918.2	P16070-13	[57]
	NCBI 35 CRA_b	425		NP_001427279.1 EAW68145.1	-	[63]
**Exons V2–V9** p.223_536delinsN	CD44v10 CD44R2 NCBI 6 UniProt 11	429		NP_001189484.1 KAI2559392.1 KAI4070742.1	P16070-11	[62]
**Exons V2–V10** p.223_604delinsR	CD44s (standard) CDw44 reticulocyte CRA_a CD44H haematopoietic NCBI 4 UniProt 12	361		NP_001001391.1 EAW68144.1 AAB13624.1 KAI2559393.1 KAI4070735.1	P16070-12	[11]
	unnamed isoform	361	p.S109Y	AAA51950.1	-	[64]
	unnamed isoform	361	p.S697I	AAH67348.1 AAM50041.1	-	[54,65]
	unnamed isoform	361	p.H92Q	AXZ96474.1	-	[66]
**Exons V2–V10** p.223_604delinsR **Exon 19** p.675_742delinsS	CD44st (short-tail) Hermes NCBI 8 UniProt 15	294		NP_001189486.1 AAB13622.1	P16070-15 H0Y5E4	[9]
**Exons V2–15** p.223_625delinsR	CD44s-exon15 NCBI 7 UniProt 18	340		NP_001189485.1 KAI2559394.1 KAI4070743.1	P16070-18	[67]
**Part of exon V3** p.266_273delinsA	UniProt 5	734	-		P16070-5	(?)
**Part of exon V3** p.G266_S273del **Exon V6** p.385_428delinsT	CRA_c UniProt 17	691		EAW68146.1	P16070-17	[59]
	unnamed isoform	691	p.I479T	AAB13625.1		[47,52]
**Exon V6** p.385_428delinsT	UniProt 6	699			P16070-6	(?)
**Exon V9** p.506_535delinsR	UniProt 7	713	-		P16070-7	(?)
**Exon V10** p.536_604delinsR	UniProt 8	674			P16070-8	(?)
**Exon 19** p.675_742delinsS	UniProt 9	675			P16070-9	(?)
	unnamed isoform	675	p.K417R p.I479T	AAB13623.1		[47,52]

**Table 2 ijms-26-09886-t002:** Isoform classification guideline for CD44 variants, following the HGVS nomenclature recommendations. For each mutation type, “p.” indicates the change occurring in the protein sequence.

Mutation Type	Example	Explanation Based on the Example
Deletion–insertion	p.385_428delinsT	Deletion of amino acids located at positions 385–428 in the canonical sequence and insertion of threonine in place of the deleted residues (between positions 384 and 429).
Single residue deletion	p.A428del	Deletion of alanine located at position 428 in the canonical sequence.
Multiple residue deletion	p.G266_S273del	Deletion of amino acids located on positions 266–273 in the canonical sequence. The first deleted residue is glycine, and the last is serine.
Single residue insertion	p.345_346insA	Insertion of an additional alanine between the residues occupying positions 345 and 346 in the canonical sequence.
Single residue substitution	p.K417R	Substitution of lysine at position 417 in the canonical sequence for arginine.

Overall, 37 groups of isoforms were distinguished based on exon composition (Figure 4). Within each group, the respective sequences varied slightly, mainly by one amino acid substitution, insertion, or deletion. Common differences include p.K417R and p.I479T substitutions classified as benign missense by UniProt; deletion of one Ala at position 428—a part of the 427–429 AAA region; and insertion of a single Ala molecule between 345 and 346 positions. Small divergences could be attributed to sequencing errors and/or natural variance between humans, and are unlikely to affect the function of the CD44 protein. We therefore put it up for discussion whether the CD44 isoforms should be distinguished based on exon content only, and that the currently available highly similar entries (i.e., NCBI 20 and NCBI 21) should be combined. Moreover, regarding the issue of isoform nomenclature, which was previously raised by Azevedo et al., 2018, we agree that standardization is needed across all databases [49]. Azevedo et al. have proposed a naming system based on the inclusion of variable exons (i.e., CD44v2-10 for the full-length form), and short abbreviations used to describe modifications outside the variable region (i.e., CD44s-exon15 (=standard form minus exon 15) for isoform NCBI 7; NP_001189485.1). At the time, only eight biological CD44 variants were listed, which made such a solution possible without overcomplicating the abbreviations. Despite its shortcomings, the Azevedo et al. naming system could serve as a basis for developing a standardized nomenclature.

Another interesting observation is that there might exist additional splice sites within exons 5 and V3 (Figure 5). Five of the examined sequences lacked the 192–223 aa located on exon 5, and four were found to be missing the final residue (266Gly) of exon 6 and 267–273 aa part of exon 7 (V3). Such structural changes occur in recognized isoforms such as NCBI 11, UniProt 3, UniProt 5, UniProt 16, and UniProt 17; however, to the best of our knowledge, they have not been described in the literature. As previously mentioned, we were unable to find source publications for the UniProt sequences. The remaining NCBI sequences come from (1) the first gene exon annotation by Screaton et al. based on the gene sequence from U.K. Medical Research Council Human Genome Mapping Resource Centre (AAB13626.1 and AAB13625.1), (2) annotation of the Celera Genomics WGS human genome (EAW68153.1 and EAW68146.1), or (3) annotation of the Genome Reference Consortium human chromosome 11 (NP_991427255.1). Annotations of the mentioned sequences yielded both isoforms containing the full exons 5, V2, and V3, as well as the showcased microexon variants. Examination of the genomic sequence (according to the most recent version of the Genome Reference Consortium genome) indicates the presence of alternative splice acceptors AG (microexon V3) or splice donors (microexons 5 and V2), which compete with the conventional splice sites, and suggests exons 5-V3 could be classified as cassette exons (Appendix A). Furthermore, in the sequences, microexon V3 is always preceded by the shorter V2 version, which could be attributed to coupling of the two splicings, or to strengthening of both V2 and V3 alternative splice sites at similar cellular conditions (i.e., presence of exonic splicing enhancers (ESEs) or exonic silencers (ESSs) [68]). These bioinformatic findings should be confirmed experimentally. Obtaining a better understanding of the splicing patterns and their regulatory factors could yield further insights into the functioning of CD44 molecules across different tissues and cancerous cells.

Apart from the variable region, the cytoplasmic domain also showed more variety than previously indicated. As mentioned in the introduction, the CD44 protein was reported to have either a short or long tail (encoded by, respectively, exon 18 or 19); however, this creates two problems. First, within the examined sequences, several contained C-terminus parts do not match either of the tail versions (Table 3). Two of the unaligned endings come from very small variants (CD44sol and CD44SP), and three from computationally annotated chromosome 11 sequences (NCBI 36, 37, X22, 38). Whether these differences arise from sequencing error or if alternate CD44 C-termini exist remains unknown, and should be further investigated. Second, since exon 18 does not contain any coding amino acids (only a distinct 3′ UTR), it can be difficult to verify its existence within a sequence. The first mention of exon 18 comes from the paper by Screaton et al., 1992 [47]—the first team to propose an exon map for the CD44 protein. That being said, we do not understand why an exon consisting of non-coding DNA only was introduced. To our knowledge, this issue has not yet been disputed.

Lastly, among the compared protein alignment, all three tools (Clustal Omega, BLASTp, and COBALT) performed equally well for sequences containing small to medium-size gaps (i.e., Figure 5); however, for larger multi-exon gaps and for partial sequences, the multiple sequence alignment tools (Figure 6A,C) were significantly more accurate than local alignment (BLASTp) (Figure 6B). The Clustal Omega software (version 1.2.4) occasionally made minor errors, especially immediately before a spliced-out exon (Figure 5A). COBALT, a constraint-based multiple alignment tool, performed the best and allowed for a more precise scoring parameter manipulation compared to Clustal Omega.

### 2.2. Structural Analysis of Experimental Three-Dimensional Structures of CD44 LinkDomain

Efforts to better understand the nature of CD44-HA signalling have also been made using a structural biology approach. The three-dimensional (3D) structures of CD44 were determined using several methods, including NMR and X-ray crystallography (Table 4). Currently, 51 murine and five human hyaluronan-binding domain (HABD; also referred to as Link module) crystal models with or without ligands can be found in the Protein Data Bank (PDB) database. Additionally, fragments of the CD44’s ICD can also be found in complexes with radixin FERM domain for both murine (PDB 2ZPY) [69] and human versions of the receptor protein (PDB 6TXS) [70]. The first two structures of human CD44 Link module were proposed by Teriete et al. [71]. The researchers assembled and compared two models—one obtained using NMR (PDB 1POZ) and the other using X-ray crystallography (PDB 1UUH, Table 4).

The ligand-binding domains are located on the ectodomain, mainly in the common region of the receptor. As previously stated, HA is the primary classical ligand of CD44. The HABD/Link domain is found in all CD44 isoforms [16] and consists of four annotated HA-binding sites: Arg-41, Arg-78, Tyr-79, and Tyr-105 (P16070 entry [72]). Other well-known CD44 ligands include OPN, collagens, MMPs [73], as well as growth factors and cytokines [48]. In contrast to HA, only selected CD44 variants (e.g., containing V6) can interact with OPN [16]. Since malignant cells have a longer variable region with more binding sites, they may transmit additional cellular signals, dysregulating the processes controlled by the receptor.

CD44’s HABD is typically reported to consist of two α-helices, as well as two triple-stranded antiparallel and four additional β-sheets [71,74] (Figure 7). There are, however, some discrepancies. In the models crystallized by Liu & Finzel [75] (hexagonal, in an unbound state—PDB 4PZ4, and monoclinic, in a complex with the unidentified co-purified peptide—PDB 4PZ3), a small third helix is present. In the sequence annotations available via the PDB, up to four α-coils can be found when examining the previously shown six proteins (Figure 7). The NMR analyses of unbound and HA-bound proteins run by Takeda et al. indicated a rearrangement of parallel β-sheet formation located near the C-terminal [76]. The resultant 3D model (PDB 2I83) presents the CD44 HABD in its ligand-bound stage; however, due to the limitations of the NMR, it was impossible to observe the oligosaccharide.

**Table 4 ijms-26-09886-t004:** Experimental structures of human and murine CD44 from the PDB. *—unpublished work.

Protein, UniProt	Amino Acids	Ligand	PDB ID	Method	Title in PDB	Ref.
**Human CD44, P16070**	20–178	-	1UUH	X-ray	Hyaluronan-binding domain of human CD44	[71]
	20–178	-	1POZ	NMR	Solution structure of the hyaluronan-binding domain of human CD44	[71]
	21–178	Bound state/no HA visible	2I83	NMR	Hyaluronan-binding domain of CD44 in its ligand-bound form	[76]
	18–170	Undefined peptide	4PZ3	X-ray	High-resolution crystal structure of the human CD44 hyaluronan-binding domain complex with undefined peptides	[75]
	18–171	-	4PZ4	X-ray	High-resolution crystal structure of the human CD44 hyaluronan-binding domain in a new space group	[75]
	678–685	ERM (ezrin/ radixin)	6TXS	X-ray	The structure of the FERM domain and helical linker of human moesin bound to a CD44 peptide	[70]
**Murine CD44, P15379**	23–174	HA8	2JCQ	X-ray	The hyaluronan-binding domain of murine CD44 in a type A complex with an HA 8-mer	[77]
	23–174	HA8	2JCR	X-ray	The hyaluronan-binding domain of murine CD44 in a type B complex with an HA 8-mer	[77]
	23–174	-	2JCP	X-ray	The hyaluronan-binding domain of murine CD44	[77]
	708–727	ERM (ezrin/ radxin)	2ZPY	X-ray	Crystal structure of the mouse radxin FERM domain complexed with the mouse CD44 cytoplasmic peptide	[78]
	23–171	HA4 or small molecules	4MRD-H, 4NP2-3	X-ray	Crystal structure of the murine CD44 hyaluronan-binding domain complex with a small molecule	[79]
	21–171	Small molecules	5BZC-Z	X-ray	Crystal structure of the murine CD44 hyaluronan-binding domain complex with a small molecule	[80] *
	21–171	Small molecules	5SBK-Z, 5SC0-7	X-ray	CD44 PanDDA analysis group deposition - The hyaluronan-binding domain of CD44 in complex with Z2856434899	[81] *

**Figure 7 ijms-26-09886-f007:**
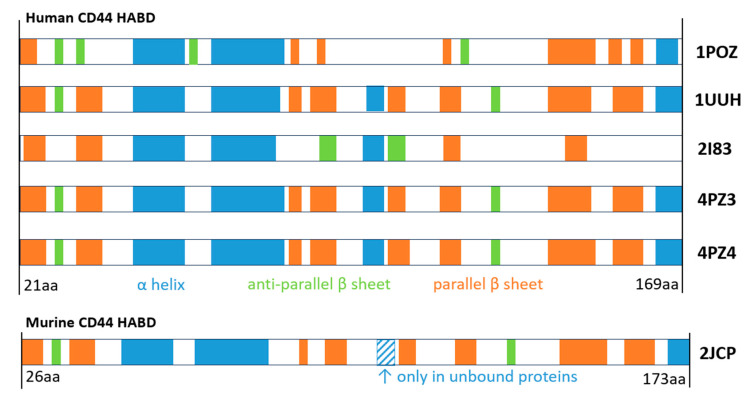
Secondary structures present in the three-dimensional models of experimental CD44 receptor Link domains—PDB IDs: 1POZ—human, unbound protein, NMR structure; 1UUH—human, unbound protein, X-ray structure; 2I83—human protein in its HA-bound state, without ligand, NMR structure; 4PZ3—human protein, bound to unknown peptides, X-ray structure; 4PZ4—human, unbound protein, X-ray structure; 2JCQ—murine protein, bound to HA octamer in a type A complex, X-ray structure. All visualizations were obtained from the PDB [82].

To date, a complete CD44-HA complex has been fully defined only for murine proteins (PDB 2JCQ and 2JCR; Figure 8), which show approximately 86% similarity with human ones. In the first murine CD44-HA complexes, more than 10 amino acid residues have been indicated as contact points for hyaluronan binding (Figure 8) [77]. In detail, the amino acids Tyr46, Cys81, Arg82, Tyr83, Ile92, Ile100, Cys101, Ala102, Ala103, Tyr109, and the disulfide bond between Cys81 and Cys101 are of particular importance in the binding of HA to the CD44 groove. Non-covalent bonds, including hydrogen bonds derived from polar residues (especially tyrosine) and hydrophobic interactions, are implicated in the attachment of this GAG to the protein molecule. It has been proven that the disulfide bond between Cys81 and Cys101 plays a crucial role in the binding of HA to CD44. It was determined that the rupture was the causative factor that impeded the binding of the polysaccharide [83]. Only five sugar rings of a hyaluronan octamer (or longer polysaccharide) interact with the murine protein, which differs from the mechanism for other HABD-containing molecules [74]. Despite the Link module being highly conserved, every HA receptor requires individual study.

Due to its involvement in numerous pathophysiological processes, CD44 has become a target protein with numerous attempts to design small-molecule binders. To date, two research groups have delivered a series of compounds characterized by X-ray cocrystal structures with murine CD44 (Table 4, chemical structures of small molecules presented in the Appendix A). Firstly, Liu & Finzel published the results of a fragment combination and structure-driven design describing a series of 1,2,3,4-tetrahydroisoquinolines as the first nonglycosidic inhibitors of the CD44–HA interaction [79]. This scaffold was further explored to yield an unpublished series of X-ray structures with PDB IDs 5BZC-Z [81]. Finally, Bradshaw et al. deposited a wide array of unpublished CD44 binders from the PanDDA analysis group deposition (PDB IDs 5SBK-Z, 5SC0-7) [80].

The first published and structurally analyzed non-glycosidic inhibitors of the CD44–HA interaction binding in an inducible pocket under the HA-binding groove were released by Liu & Finzel in 2014 (Figure 9). Iterations of fragment combination and structure-driven design have allowed identification of a series of 1,2,3,4-tetrahydroisoquinoline pharmacophores as an attractive starting point for lead optimization. Among six deposited cocrystal structures, the two most elongated molecules (PDB IDs 4NP2 and 4NP3), which already stretch into the HA-binding groove, could be developed to create more profound inhibition [79].

The tetrahydroisoquinoline scaffold was further advanced by the same research group, which resulted in an unpublished series of murine CD44/small-molecule inhibitor cocrystal structures released in 2016 (PDB IDs 5BZC-Z, Figure 10). In this study, the majority of compounds were binding as expected to the pocket neighbouring the HA groove, additionally augmenting it by the interacting tetrahydroisoquinoline core. Interestingly, one compound bound only in the new bottom pocket (2-(2-methoxyethyl)-1,2,3,4-tetrahydroisoquinolin-8-amine from 5BZN), and one in both the HA groove and bottom pockets (2-(1,3-dimethoxypropan-2-yl)-1,2,3,4-tetrahydroisoquinolin-8-amine from 5BZT). Among the compounds which form contacts at the very centre of the HA-binding site, a six-membered heterocyclic aliphatic ring was a necessary substituent (as found in 2-[3-(tetrahydro-2H-pyran-4-yloxy)propyl]-1,2,3,4-tetrahydroisoquinolin-8-amine from 5BZR, 2-[3-(tetrahydro-2H-pyran-4-yloxy)propyl]-1,2,3,4-tetrahydroisoquinolin-5-amine from 5BZS, and 2-[3-(morpholin-4-yl)propyl]-1,2,3,4-tetrahydroisoquinolin-8-amine from 5BZQ). Although this is an unpublished series of compounds with no experimental binding assay results provided, it seems highly probable that these optimized molecules constitute a group of efficient competitive CD44-HA interaction inhibitors. There were also other approaches to develop this first-published novel scaffold, e.g., by in silico design in 2021, reporting two hits (Can125 and Can129) with high theoretical affinity to the murine and human structures of CD44 HABD [84].

Concerning the most recent structurally characterized series of compounds, Bradshaw et al. released, in 2021, a wide array of unpublished CD44 binders from the PanDDA analysis group deposition (PDB IDs 5SBK-Z, and 5SC0-7) [80]. As a result of this fragment-based assay, five pockets (1–5) have been identified on the surface of murine CD44 (Figure 11). In pocket 1 (analogical to the bottom pocket of Liu & Finzel, 2016 unpublished series [81]), Z2856434874 (5SBW), Z2856434899 (5SC0), and Z2856434878 (5SBO) molecules have been observed to bind with the involvement of Pro147 in a small groove located on the opposite side of the protein from the HA-binding groove. Moving to pocket 2, the Z126932614 (5SBL) molecule has been observed to bind in the proximity of the region for Z2856434874, Z2856434899, and Z2856434878, with the participation of Val48, Ile26, Asn43, and Lys42. It is noteworthy that this is the region where the HABD domain has its N- and C-terminal ends. Therefore, under native conditions, when the entire CD44 protein is expressed, it is predicted that this ligand is unlikely to bind. The binding sites for Z445856640 (5SBT) and POB0120 (5SC7) have been shown to engage Asn29 in pocket 3. It has been established that these compounds are located in proximity to the HA-binding groove. Therefore, they should be considered as potential inhibitors of this GAG binding, due to the steric hindrance they may create for such a large ligand. The Z1267885772 (5SBM) molecule binds with the participation of Tyr34 and Ile95 at a negatively charged surface in proximity to the HA-binding groove (pocket 4); however, its presence is likely not to affect the binding of this polysaccharide. Other ligands, i.e., Z1258992717 (5SBK), Z57040482 (5SBN), Z1229798311 (5SBP), Z44592329 (5SBQ), Z1259341012 (5SBR), Z340495298 (5SBS), Z839988838 (5SBU), Z31721798 (5SBV), Z53825479 (5SBX), Z1878656559 (5SBY), Z768399682 (5SBZ), Z431807512 (5SC1), and Z190780124 (5SC2) bind in pocket 5, located on the opposite side of CD44 from the HA-binding groove (analogical to the undefined peptide pocket of Liu & Finzel, 2014, [75]). The amino acids involved in the binding of these ligands (depending on the structure of the small molecule compound) are Arg33, Phe38, Phe60, Asn125, Asp133, Ser136, Val137, Thr138, Asp139, and Pro141. Ligands that bind by engaging Asn125 most frequently tilt it from the conformation present in the structure only with the HA (PDB ID 2JCQ).

It was observed that none of the ligands described in the PanDDA library caused conformational differences within the HA groove that would certainly inhibit its ability to bind. In the HA-binding groove, the presence of PanDDA compounds resulted in only minor alterations, characterized by subtle deviations of individual atoms. The amino acid Arg82 and hydrogens from the –OH groups of Tyr83 and Tyr109 were most frequently affected by these deviations. These modifications did not cause any deformation of the HA-binding site.

Although the disulfide bond between Cys81 and Cys101 was intact, the sulphur atoms were not always oriented in the same direction as in the structure of murine CD44 with HA (PDB ID 2JCQ), resulting in its slight shift from the described GAG. Nevertheless, these alterations should not be regarded as having the potential to impede polysaccharide binding. It has been established that both cysteines are located on loops that are characterized by a certain flexibility. This means that the disulfide bond has the potential to be in a position that allows it to interact with HA.

The third pocket, located in proximity to the HA-binding groove, has been observed to undergo conformational changes in response to the presence of small molecules. Due to its location, ligands attached at this site may affect the binding of HA to CD44. One of the molecules under study, Z445856640 (5SBT), has been observed to bind to the surface of the protein without inducing alterations in the allosteric pocket. In contrast, the POB0120 (5SC7) molecule has been found to induce conformational changes that “lengthen” the binding pocket, thus allowing it to accommodate the molecule.

The only amino acids that underwent conformational changes were those absent in the HA-binding groove, Arg155, Arg45, and Arg50. However, it should be noted that arginine is an amino acid with a relatively long carbon chain in its side chain, which allows it to bend and rotate in multiple directions, depending on external factors.

In summary, the structures of murine CD44 in complex with HA octamer revealed that the binding site for HA forms a shallow groove dominated by hydrogen bonds, which is not an ideal situation for binding small molecules. However, the opening of a small binding pocket adjacent to the HA groove was found to be induced by the binding of 1,2,3,4-tetrahydroisoquinoline derivatives, as demonstrated by Liu & Finzel, 2014 [79], and two series of deposited cocrystal structures (Liu & Finzel, 2016, and Bradshaw et al., 2021, unpublished) [80,81]. The tetrahydroisoquinoline scaffold also served as the basis for the in silico development of a series of CD44 antagonist candidates, which can be easily synthesized by convenient multicomponent reactions [84]. Several studies have attempted to discover novel direct binders to human, not murine CD44, using NMR studies [85,86]. In 2016, Baggio et al. [85] published the results of a fragment-screening campaign, in which the best compound 131B6 bound to the target Link domain with an affinity of 7.5 mM [85]. Pustuła et al., 2019, found four initial active structures by high-throughput screening of a commercially available library of small molecules [86]. The structures (Appendix A) were to some extent similar to 1,3-thiazole derivatives previously described by Baggio et al. [85], however, the reported binding was approximately one order of magnitude higher (between 0.66 mM and 2.65 mM). Despite the valuable input of providing the structures of human CD44 binders based on cores distinct from tetrahydroisoquinoline, unfortunately, none of the NMR-characterized molecules were shown to bind in proximity to the HA-binding groove, affecting the residues at the other side of this pocket. In the case of compounds 1, 2, and 3, the most significant chemical shift perturbations are observed for Gly159, Asp128, Leu135, and Asp134 (notably for compound 3). For the hit compound 131B6, Gly159, Asp134, Leu135, Ile145, Ile147, Ala138, and Lys158 were the most prominently impacted [85,86].

Finally, to further explore the druggability of the CD44 surface, especially the human one in reference to its better characterized murine counterpart, we performed the hot spot analysis using the FTMap server (Figure 12). To identify putative druggable binding pockets, sixteen chemically diverse small-molecule probes are docked onto the investigated protein’s surface by FTMap. The primary hot spots, also referred to as consensus sites, were those with the highest numbers of bound probe clusters (labelled as cyan, Figure 12A,B), followed by secondary (magenta), tertiary (yellow), and quaternary (pale pink) hot spots. Comparing the outcomes for human (Figure 12A, PDB ID 1UUH) and murine (Figure 12B, PDB ID 2JCP) CD44, interestingly, the four most druggable pockets are localized in similar locations of the CD44 surface, which may indicate high structural similarity of these homologues. Most of the probe clusters bind onto or in the proximity of the broad HA-binding groove (Figure 12D, a close-up view for murine CD44). The main difference between these two instances is that the primary hot spot (cyan) for the human receptor is located in the HA-binding groove, whereas for the murine analogue, it is in the so-called “peptide” or “5” pocket, distant from the HA-site. This well-defined cleft was a site targeted in both human (a close-up view comparing yellow tertiary hot spot on green human CD44 with an undefined peptide and one Liu&Finzel compound, Figure 12C, see also Figure 9A) and murine CD44 (a close-up view comparing cyan primary hot spot on grey murine CD44 with numerous PanDDA compounds, Figure 12C, see also Figure 11B). Concerning the probe clusters for the well-explored murine HA-binding groove, secondary, tertiary and quaternary hot spots, along with other less druggable pockets (e.g., hot spot no. 7, orange), overlap with the HA octamer or tetrahydroisoquinoline Liu&Finzel compounds, corroborating the possible utility of this scaffold also for the human CD44 (secondary magenta hot spots on both homologues; a close-up view comparing several hot spots on grey murine CD44 with numerous tetrahydroisoquinoline compounds, Figure 12D, see also Figure 10C). Despite earlier classification of CD44s, a hardly druggable receptor, the presented outcomes suggest the possibilities of small-molecule binder discovery based on various chemistries, in contrast to really difficult cases, such as the PD1-PD-L1 complex, characterized by a flat interface of interactions. Taken together, the hot spot analysis by the FTMap server opens up a prospect for further exploration of the human CD44 surface for targeting of both the broad HA-binding groove in which several hot spots are located, but also the auxiliary, small pocket distant from the HA-site, which is proven to bind, e.g., peptides or numerous small molecules in the case of murine CD44.

In parallel to small molecules, other strategies have been developed to target CD44 and its ligands. For example, neutralizing antibodies against CD44 (Bivatuzumab, KM201, U36, VF18, and RG7356) are in development and are at various stages of clinical trials [87]. Monoclonal antibody RG7356 was evaluated in 65 patients with advanced CD44-expressing solid malignancies in the first-in-human phase I study (ClinicalTrials.gov Identifier NCT01358903) sponsored by F. Hoffmann-La Roche Ltd., Basel, Switzerland [88]. RG7356, a monoclonal antibody targeted to the constant region of CD44, showed an acceptable safety profile. The study was terminated early due to the lack of evidence of a clinical and/or pharmacodynamic (PD) dose–response relationship with RG7356, but not due to safety concerns. Consequently, the optimal biological dose schedule was not achieved. Tumour targeting from doses ≥200 mg warrants further investigation of RG7356, perhaps in combination regimens. New clinical approaches to CD44 drugging are also highlighted, for example, by the study of Liu et al., 2023, which shows that CD44 is a potential immunotherapeutic target affecting macrophage infiltration and leading to a poor prognosis [87]. The expression of CD44 may be related to the immune escape of tumour cells through a regulatory effect on immune checkpoint genes, e.g., PD-L1 in bladder cancer.

Other approaches utilized in CD44 targeting include short, single-stranded DNA aptamers. For instance, multiple rounds of in vitro selection led to the development of CD44 HABD-recognizing nanomolar-affinity aptamers, which efficiently inhibited the growth of leukemic cancer cells characterized by high CD44 expression [89]. Further preclinical development should focus on optimizing molecular size by identification of the minimal binding region, elucidating the involved molecular mechanisms in more detail, conjugating aptamers, and assessing in vivo activity. Taken together, the aptamers broaden the existing landscape of potential approaches to the development of antitumor strategies based on inhibition of the CD44 axis.

Concerning the structural analysis of the intracellular part of CD44, despite no defined fold of the cytoplasmic region (as visible on, e.g., AlphaFold ID AF-P16070-F1), transient protein folding of this fragment was demonstrated by two X-ray structures of FERM (four-point-one/ezrin/radixin/moesin) complexes with CD44-derived peptides (Figure 13). Chronologically, in 2008, Mori et al. released the crystal structure of the mouse radixin FERM domain complexed with the mouse CD44 cytoplasmic peptide (PDB ID 2ZPY), published in the same year [78]. Following the rodent model, in 2020, the structure of the FERM domain and helical linker of human moesin bound to a CD44 peptide was released (PDB ID 6TXS), which was further described in a publication of Du et al., 2023 [70]. In both cases, the fragment of CD44 cytoplasmic domain folds into a β-strand, which binds the shallow groove between strand β5C and helix α1C and augments the β-sheet β5C-β7C from subdomain C of murine FERM. From a functional point of view, ERM family proteins connect the actin cytoskeleton to the plasma membrane, thereby regulating the structure and function of specific domains of the cell cortex. Concerning the CD44/moesin (MSN) interaction, proteomic studies have identified FERM-containing MSN and the receptor CD44 as hub proteins found within a co-expression module strongly linked to Alzheimer’s disease (AD) traits and microglia. Both proteins are more abundant in the brains of AD patients, and their levels are positively correlated with amyloid plaque deposition, cognitive decline, and neurofibrillary tangle burden. Inhibiting the MSN-CD44 interaction may help limit AD-associated neuronal damage [70].

## 3. Materials and Methods

### 3.1. Data Source Selection

The human protein sequences were obtained from UniProt and the NCBI databases. The UniProt P16070 CD44_Human entry included 19 isoform sequences and 19 computationally mapped potential variants. Within the NCBI database, a search using the terms “*Homo sapiens*” as organism, and “CD44” as either gene name or protein name was performed and yielded 106 results, of which 7 were excluded: 6 very short protein chains (A, B, BBB) and 1 non-human sequence (*Pongo abelii*). Additionally, for the remaining 81 complete and 18 partial NCBI entries, all available structures listed by the “Identical protein” feature were also considered.

### 3.2. Sequence Alignments in Clustal Omega, BLASTp, and COBALT

The selected sequences were first grouped by size (± 1 amino acid residue) under the assumption that highly similar sequences would also be similar in size. Each group was then aligned to the full-length canonical CD44 variant (Table 5, NP_000601.3, P16070-1) using the Clustal Omega Multiple Sequence Alignment tool for protein sequences [90] (ClustalW with character counts version 1.2.4); BLASTp local alignment programme version 2.17.0 [91,92] (expect threshold 0.05; word size 3; max matches in a query range 0; matrix BLOSUM45; Existence: 19; Extension: 1; conditional compositional score matrix adjustment); and COBALT [93] version 1.26.0 (gap penalties: opening -11, extension -1; end-gap penalties: opening -5, extension -1; RPS blast: yes; E-value constrain 0.005; conserved columns: yes; query clustering: yes; word size 4; max cluster distance 0/8; alphabet SE-B15). Significantly divergent sequences with low query covers were additionally examined in one-to-one alignments to the canonical form.

### 3.3. Structural Analysis of CD44-Ligand Complexes

The PDB [82] was searched using the “CD44” query, resulting in 72 structures, out of which 14 were excluded as they did not contain the fragments of CD44 in any of their chains. Out of the remaining 58 structures, 6 were of human origin, and 52 were of mouse origin. Experimental CD44 structures, as determined by X-ray crystallography or nuclear magnetic resonance spectroscopy (NMR), were analyzed using PyMOL (version 2.5.2, Schrödinger, LLC, New York, NY, USA) or Flare^TM^ (version 10.0.0) from Cresset^®^ (Buenos Aires, Argentina). Chains A of the models were chosen for protein alignments made to analyze ligand binding modes. Chemical structures of the molecules were drawn in ChemSketch (version 2023.2.4, ACD/Labs, Toronto, ON, Canada).

### 3.4. Hot Spot Analysis of Human and Murine CD44

Hot spot detection on the CD44 surface was performed using the FTMap server available at http://ftmap.bu.edu (accessed on 3 October 2025) [94] from Boston University, USA, using standard parameters. Chain A of human and murine CD44 X-ray structures in the unbound states was analyzed with prior removal of water and other small molecules (PDB ID 1UUH and 2JCP, respectively). The list and structures of the compounds docked in FTMap can be found, e.g., in [95]. The results showing the location of probe clusters were inspected and visualized in PyMOL (version 2.5.2, Schrödinger, LLC). The recommendations from previously published publications were followed during the interpretation and description of the FTMap data [94,95].

## 4. Conclusions

Research on the CD44 receptor, comprising approximately 40 years of studies since its discovery in 1987, has resulted in an impressive total of 5200 publications in NCBI PubMed (CD44 term searched in the title), including 1009 articles in the last 5 years and 221 in the last year. Due to its involvement in tumour metastasis, carcinogenesis, and other pathophysiological processes, numerous efforts have been made to analyze its structure and target it with antibodies, peptide mimetics, aptamers, or pharmacological compounds. Targeting CD44 with direct small-molecule inhibitors is considered to be challenging due to the lack of a well-defined, deep binding pocket. Moreover, the X-ray structure of the human CD44 Link domain with its key ligand, the HA octamer, is still missing. On the other hand, the HABD of the murine homologue is well characterized and co-crystallized with numerous ligands, including HA and a series of small-molecule binders. Several CD44-targeting compounds are already optimized versions of the 1,2,3,4-tetrahydroisoquinoline fragment that binds in the proximity of the HA-binding groove. It should, however, be noted that even despite the structural similarity of murine and human homologues, all potential competitive inhibitors of HA binding should be validated in assays with the human CD44.

Concerning the bioinformatic analysis of published or deposited CD44 sequences, several obstacles were encountered during the attempts to systematize the existing CD44 variants. First of all, some publications lack a direct reference to a concrete sequence accession number and are described only by specifying the exons contained in the protein. The division into 19 exons was made in 1992 by Screaton et al. [47], shortly after the discovery of CD44. In the majority of the exons, their sequences can be clearly identified in the analyzed variants; however, the situation with others is ambiguous, particularly with poorly defined exon 18 or exons 5/7, where the alternative endings exist in repetitive variants. Secondly, mutations are also present within the deposited variants, out of which some are described as benign; most, however, lack a comment on their biological significance. Finally, certain variants are not identified by one of the protein sequence databases (NCBI Protein/UniProt), and an increasing number of predicted variants are being successively proposed by the analyses of updated human genome sequencing efforts.

The situation of a lack of order is not very surprising, since knowledge about the receptor was accumulating rapidly after its identification as being involved in cancer. An ideal situation for the CD44 community would be one in which all variants have their accession numbers assigned in both the NCBI Protein and UniProt databases. We would like to leave the reader with the recommendation to specify both NCBI and UniProt accession numbers and/or present the alignments of the investigated variant to the canonical CD44 sequence in upcoming publications. This inconsistency in CD44 variants classification is of particular concern because different isoforms of CD44 are known to mediate distinct biological functions, including regulation of cell adhesion, migration, proliferation, and immune response. Therefore, accurate annotation of isoforms is not only an academic exercise but also a prerequisite for translating CD44-related findings into clinical applications.

## Figures and Tables

**Figure 3 ijms-26-09886-f003:**
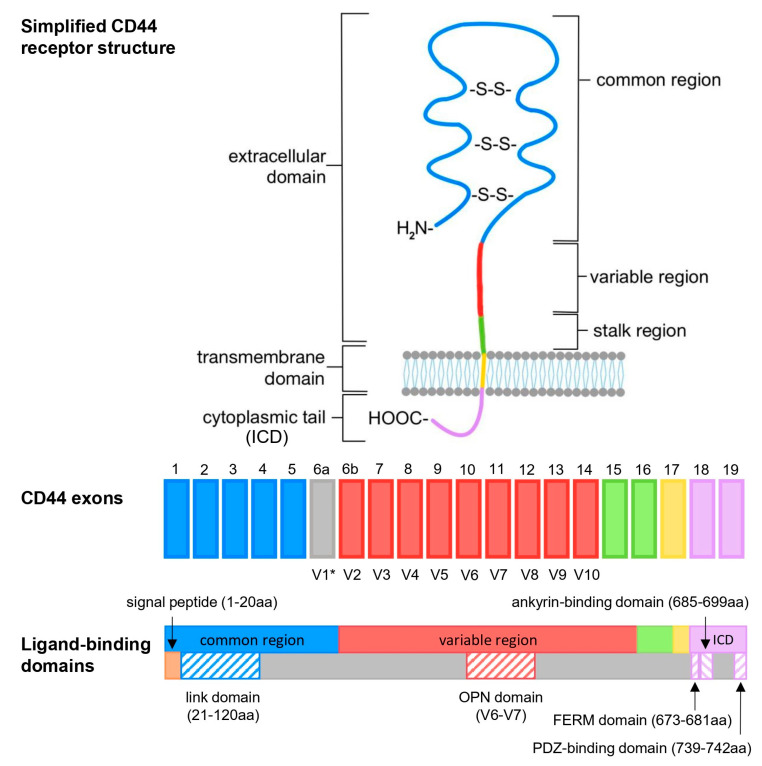
Schematic structure of the CD44 receptor and its encoding sequence. * Variable exon V1 is only found in mouse CD44, hence why it was not assigned a number in the human CD44 model. Abbrv: ICD—intracellular domain; OPN—osteopontin; FERM—4.1/ezrin/radixin/moesin; PDZ—PSD-95/Dlg/ZO-1 binding domain.

**Figure 4 ijms-26-09886-f004:**
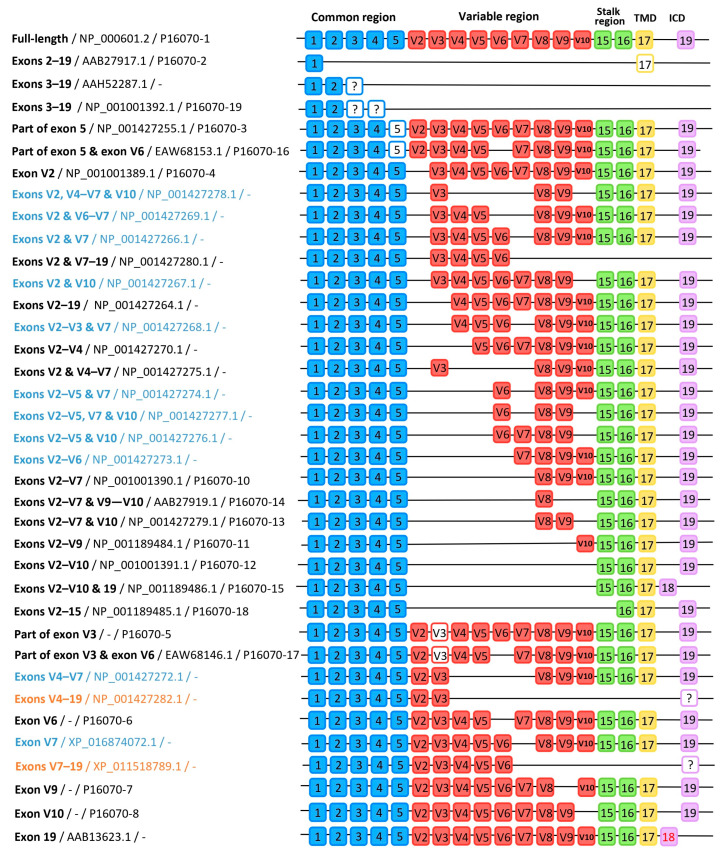
Graphical representation of the 37 exon patterns found among the examined CD44 isoforms, designated by the spliced-out (parts of) exon(s)/NCBI accession/UniProt accession, and colour-coded: black—Table 1, blue—Appendix A, orange—Table 2. The order of appearance is the same as in Table 1 and Appendix A. White square with colourful frame indicates an exon present only partially; unaligned sequence fragments are indicated with a question mark (?); the red font indicates a possibility of a short tail (based on amino acids RRS at the end of exon 17).

**Figure 5 ijms-26-09886-f005:**
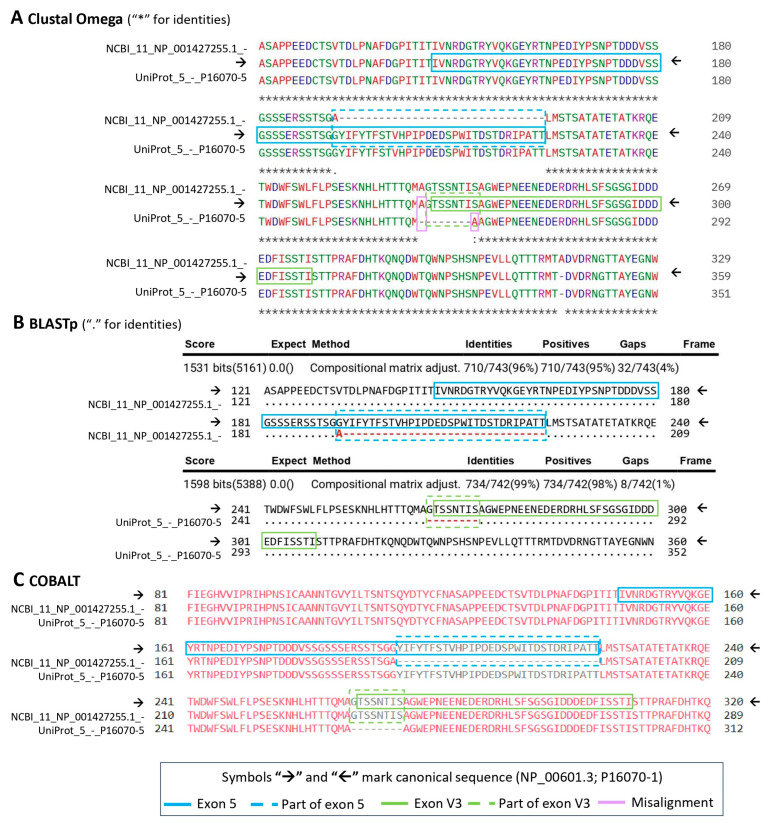
Fragments of the alignment of isoforms NCBI 11 (NP_001427255.1) and UniProt 5 (P16070-5) to the full-length CD44 sequence (NP_000603.1; P16070-1) were performed using the following: (**A**) Clustal Omega (version 1.2.4); (**B**) BLASTp suite (version 2.17.0); (**C**) COBALT (version 1.26.0); the full exons 5 and 7 (V3). These, as well as their spliced-out parts, are indicated in blue and green; misaligned residues are shown in lilac; for every method, the reference canonical sequence is signified with the arrow symbols “🡢” and “🡠”. Dash lines in the alignments indicate a deletion.

**Figure 6 ijms-26-09886-f006:**
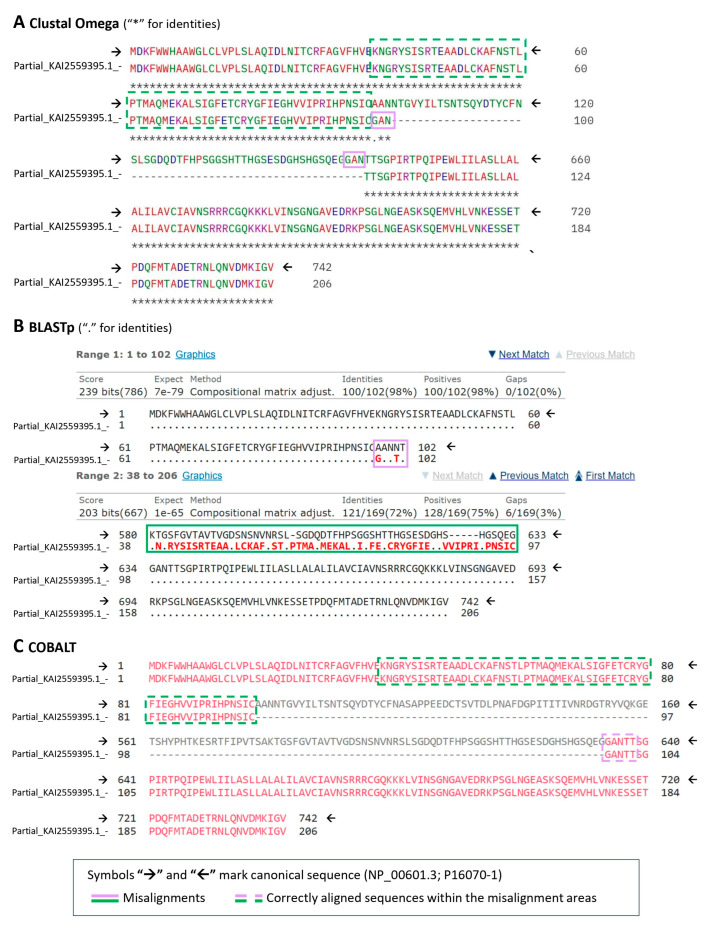
Fragments of the alignment between a partial sequence KAI2559395.1 and the CD44 canonical form made using (**A**) Clustal Omega (version 1.2.4); (**B**) BLASTp suite (version 2.17.0); (**C**) COBALT constraint-based multiple alignment tool. (version 1.26.0) Misalignments are shown in solid line lilac and dark green boxes; dashed line indicates the correct alignment in places where errors did occur; for every method, the reference canonical sequence is signified with the arrow symbols “🡢” and “🡠”.

**Figure 8 ijms-26-09886-f008:**
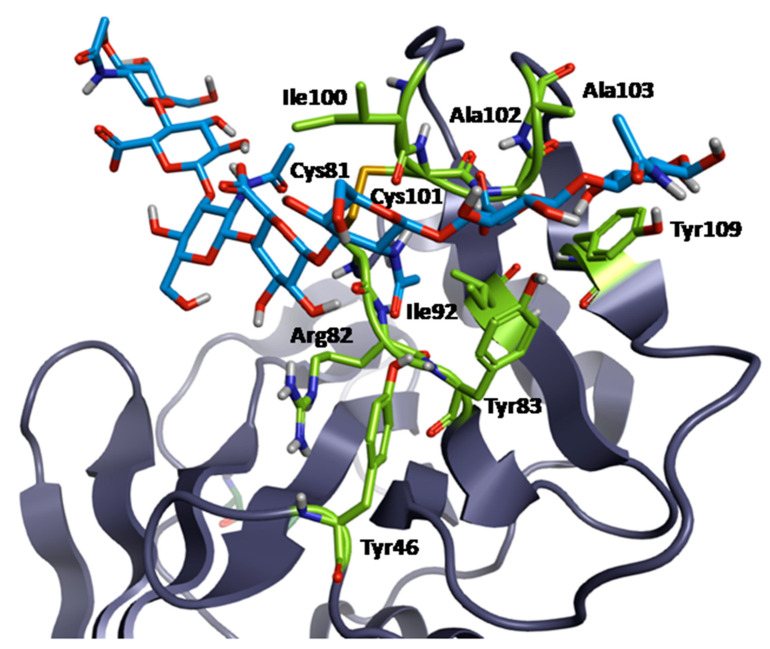
X-ray structure model of CD44-HA complex (PDB 2JCQ—murine CD44 bound to HA octamer in type A complex). Key interacting residues are depicted as green sticks with coloured heteroatoms, HA as cyan sticks with coloured heteroatoms, protein chain A (cartoon) as grey-purple cartoon. Image generated using Flare™ (version 10.0.0) from Cresset^®^.

**Figure 9 ijms-26-09886-f009:**
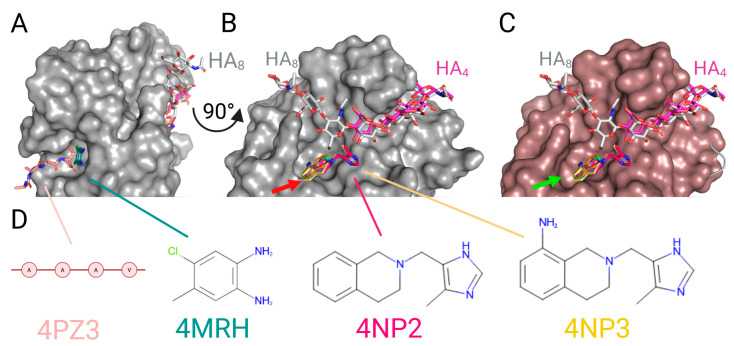
X-ray structure models of murine CD44 receptor with small molecules by Liu & Finzel, 2014 [79]. The following chains were aligned: light grey: 2JCQ (murine CD44 with HA_8_); pale pink: 4PZ3 (human CD44 with undefined peptide AAAV). Structures of murine CD44 from published Liu & Finzel series: dark magenta: 4MRD (with HA4); blue marine: 4MRE; orange: 4MRF; green: 4MRG; dark turquoise: 4MRH; reddish pink: 4NP2; yellow: 4NP3. All compounds visualized as sticks, with coloured heteroatoms. (**A**) Side view of CD44 (surface of murine CD44 2JCQ depicted in grey) visualizing two binding pockets: main pocket fusing with HA-binding groove, and back pocket where the undefined peptide was found to associate with human CD44. (**B**) 90° turn of panel A showing the front view of the HA-binding groove (CD44 grey surface from 2JCQ); all tetrahydroisoquinoline derivatives bind into the pocket under the HA chain, with two molecules stretching to the HA-binding site directly; there exists a steric clash between the molecule scaffold and the CD44 surface from 2JCQ, which is eliminated in panel C (red arrow). (**C**) The same orientation as on panel B, but with a surface of murine CD44 from 4NP2 depicted in reddish pink; the tetrahydroisoquinoline core augments the inducible pocket (green arrow) for binding of the small molecules under the HA-binding groove. (**D**) Chemical structures of the chosen molecules: 4-chloro-5-methylbenzene-1,2-diamine from 4MRH, which binds in the AAAV peptide binding pocket from 4PZ3; compounds 2-[(4-methyl-1H-imidazol-5-yl)methyl]-1,2,3,4-tetrahydroisoquinoline and 2-[(4-methyl-1H-imidazol-5-yl)methyl]-1,2,3,4-tetrahydroisoquinolin-8-amine (4NP2 and 4NP3, respectively), which constitute the competitive inhibitors of HA binding. Created with BioRender.com.

**Figure 10 ijms-26-09886-f010:**
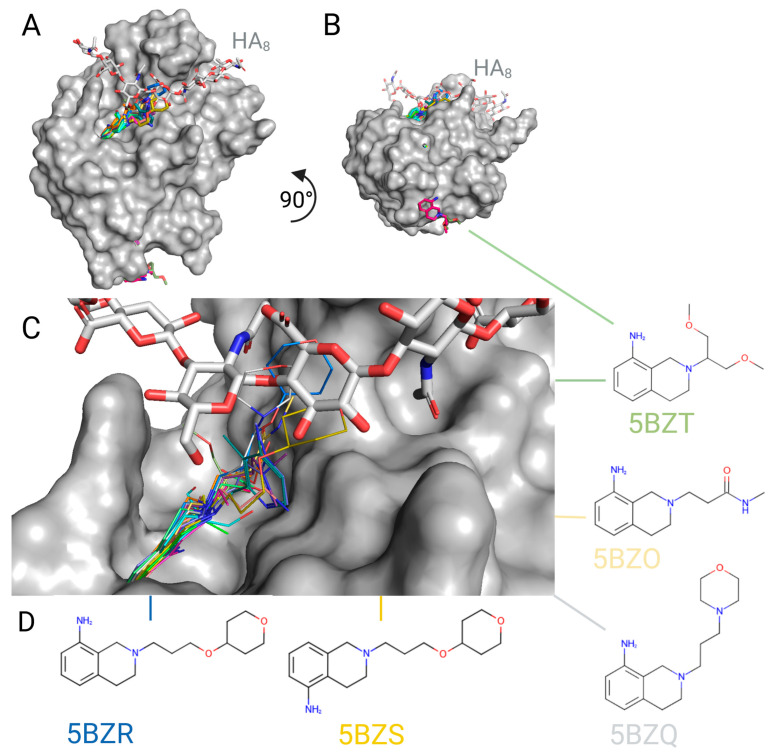
X-ray structure models of murine CD44 receptor with small molecules by Liu & Finzel, 2016, unpublished [81]. The following chains were aligned: light grey: 2JCQ (murine CD44 with HA8), with the structures of murine CD44 from unpublished Liu & Finzel series (PDB IDs 5BZC-Z), where the most significant binders in the proximity of the HA-binding groove are as follows: dark blue: 5BZR; lemon: 5BZS; grey: 5BZQ; pale yellow: 5BZO; olive: 5BZT. All compounds are visualized as sticks or lines, with coloured heteroatoms (e.g. magenta and olive in the bottom pocket, etc.). (**A**) The front view of CD44 (surface of murine CD44 2JCQ depicted in grey), visualizing the main pocket fusing with the HA-binding groove, and the back pocket, where the undefined peptide was found to associate with human CD44. (**B**) 90° turn of panel A, showing the bottom view and two binding pockets: at the bottom and at the HA-binding groove (CD44 grey surface from 2JCQ). All tetrahydroisoquinoline derivatives (except for 5BZN, dark pink, and 5BZT, olive, which bind simultaneously to both clefts) bind into the pocket under the HA chain, with several molecules stretching to the HA-binding site directly. (**C**) Close-up front view of the main pocket. (**D**) Chemical structures of the chosen molecules which compete the most with HA binding: 2-[3-(tetrahydro-2H-pyran-4-yloxy)propyl]-1,2,3,4-tetrahydroisoquinolin-8-amine from 5BZR, 2-[3-(tetrahydro-2H-pyran-4-yloxy)propyl]-1,2,3,4-tetrahydroisoquinolin-5-amine from 5BZS; 2-[3-(morpholin-4-yl)propyl]-1,2,3,4-tetrahydroisoquinolin-8-amine from 5BZQ, 3-(8-amino-3,4-dihydroisoquinolin-2(1H)-yl)-N-methylpropanamide from 5BZO, and 2-(1,3-dimethoxypropan-2-yl)-1,2,3,4-tetrahydroisoquinolin-8-amine from 5BZT. Created with BioRender.com.

**Figure 11 ijms-26-09886-f011:**
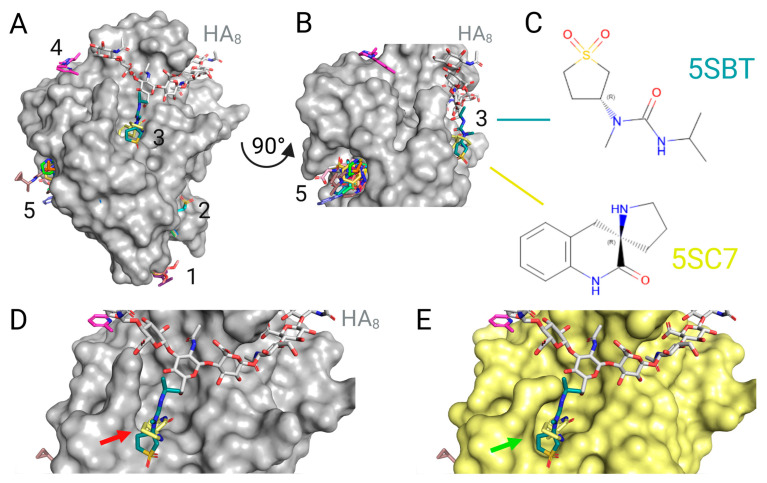
X-ray structure models of murine CD44 receptor with small molecules by Bradshaw et al., 2021, unpublished [80]. The following chains were aligned: light grey: 2JCQ (murine CD44 with HA_8_), structures of murine CD44 from the unpublished Liu & Finzel series (PDB IDs 5SBK-Z, 5SC0-7), where the most significant binders are as follows: dark marine: 5SBT; lemon: 5SC7. All compounds visualized as coloured sticks, with coloured heteroatoms (e.g. magenta in pocket no 4, dark marine and lemon in pocket no 3, etc.). (**A**) Front view of CD44 (surface of murine CD44 2JCQ depicted in grey), visualizing five binding pockets, where pocket 3 is fusing with the HA-binding groove, and back pocket 5 is the cleft where the undefined peptide from 4PZ3 was found to associate with human CD44. (**B**) 90° turn of panel A showing the side view, visualizing pockets 3 and 5. (**C**) Chemical structures of the competitive binders: Z445856640: N-[(3R)-1,1-dioxo-1lambda~6~-thiolan-3-yl]-N-methyl-N’-propan-2-ylurea from 5SBT and POB0120: (2R)-1′,4′-dihydro-2′H-spiro[pyrrolidine-2,3′-quinolin]-2′-one from 5SC7. (**D**) Close-up front view of the HA-binding groove fused with pocket 3 (CD44 grey surface from 2JCQ); compounds from 5BST and 5SC7 bind into the cleft with a slight steric clash between molecules and the CD44 surface from 2JCQ, which is eliminated in panel E (red arrow). (**E**) The same orientation as on panel (**D**), but with a surface of murine CD44 from 5SC7 depicted in yellow; (2R)-1′,4′-dihydro-2′H-spiro[pyrrolidine-2,3′-quinolin]-2′-one significantly augments the inducible pocket (green arrow) for binding of the small molecules under the HA-binding groove. Created with BioRender.com.

**Figure 12 ijms-26-09886-f012:**
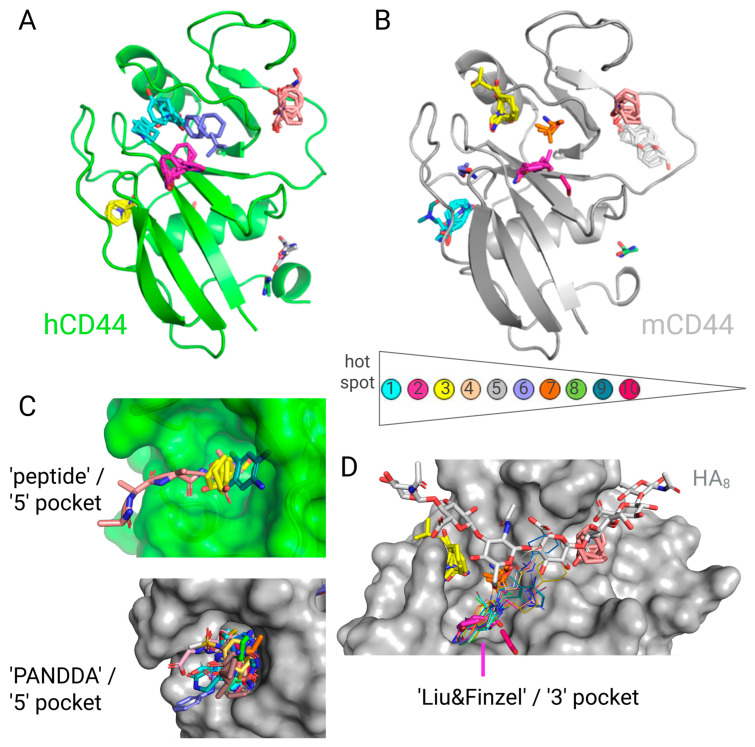
The FTMap server analysis of human (hCD44) and murine (mCD44) structures. Results for: (**A**) human (PDB ID 1UUH); (**B**) murine (PDB ID 2JCP) CD44, with the colour code for the hot spots in the legend (primary hot spots are those with the highest numbers of bound probe clusters, followed by secondary (magenta), tertiary (yellow), quaternary (pale pink), etc.; with the red hot spot no. 10 as the least druggable pocket). Close-up views: (**C**) comparing the yellow tertiary hot spot on green human CD44 with an undefined peptide (pale pink) and one Liu&Finzel compound (marine), and the cyan primary hot spot on grey murine CD44 with numerous PanDDA compounds; (**D**) comparing several hot spots (secondary, tertiary, and quaternary hot spots, along with other less druggable pockets, e.g., hot spot no. 7, orange) on grey murine CD44 with the HA octamer (light grey sticks with coloured heteroatoms), and numerous tetrahydroisoquinoline compounds shown as lines. Pocket numbering (“5” or “3”) refers to the numbering on Figure 11, where pocket 3 is in close proximity to the HA-binding groove, and pocket 5 is a distant site uncompetitive to HA binding. Created with BioRender.com.

**Figure 13 ijms-26-09886-f013:**
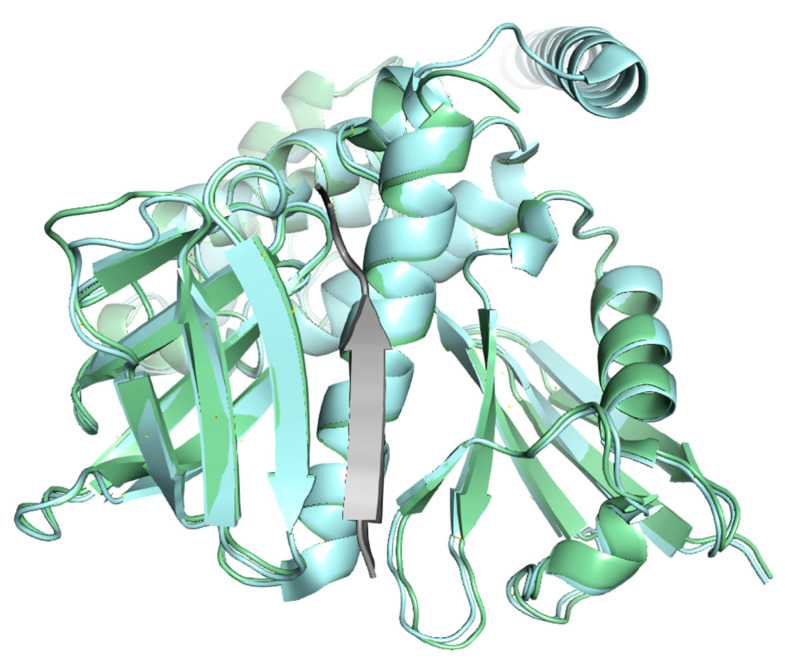
Three-dimensional models of X-ray structures of ERM (ezrin/radxin) complexes with CD44-derived peptides. The following structures were aligned: 6TXS—the structure of the FERM domain (light cyan) and helical linker of human moesin bound to a CD44 peptide (residues 678–685, light grey), and 2ZPY—crystal structure of the mouse radixin FERM domain (light green) complexed with the mouse CD44 cytoplasmic peptide (residues 708–727, dark grey). All protein chains are visualized as cartoons. In both cases, the fragments of CD44 cytoplasmic domains fold into β-strands that join with the existing antiparallel β-sheets of ERM proteins.

**Table 3 ijms-26-09886-t003:** List of alternative C-terminus endings found among the available CD44 isoform sequences.

Abbreviation(s)	Accession(s)	Full Exons Included	Unaligned Sequence C-End
CD44SP UniProt 2	P16070-2 AAB27917.1	1	GVGRRKS
CD44sol CD44RC, CRA_g NCBI 5 UniProt 19	NP_001001392.1 P16070-19 AAC70782.1 EAW68152.1 KAI2559396.1 KAI4070741.1	1–2	SLHCSQQSKKVWAEEKASDQQWQWSCGG QKCGGQKAKWTQRRGQQVSGNGAFGEQ GVVRNSRPVYDS
NCBI 36 NCBI 37	NP_001427280.1 NP_001427281.1	1–5, 7–10 (V3–V6)	GDCGSMAWVKKYFSFIFL
NCBI X22	XP_054226555.1 XP_011518789.1	1–10 (1-V6)	
NCBI 38	NP_001427282.1	1–7 (1-V3)	IICLFTRRIYKQHTVTKSLGFQVQRDTTDCMD GQNGAFGYPRWRAGVFKAVLPTAAASLTVL SGRSHVLNPKVFYDRMQRTLRCLPIWLN

**Table 5 ijms-26-09886-t005:** CD44 canonical sequence (P16070-1/NP_000601.3).

CD44 Canonical (Amino Acids 1–742)
MDKFWWHAAWGLCLVPLSLAQIDLNITCRFAGVFHVEKNGRYSISRTEAADLCKAFNSTLPTMAQMEKALSIGFETCRYGFIEG HVVIPRIHPNSICAANNTGVYILTSNTSQYDTYCFNASAPPEEDCTSVTDLPNAFDGPITITIVNRDGTRYVQKGEYRTNPEDIYPS NPTDDDVSSGSSSERSSTSGGYIFYTFSTVHPIPDEDSPWITDSTDRIPATTLMSTSATATETATKRQETWDWFSWLFLPSESKNHLH TTTQMAGTSSNTISAGWEPNEENEDERDRHLSFSGSGIDDDEDFISSTISTTPRAFDHTKQNQDWTQWNPSHSNPEVLLQTTTRM TDVDRNGTTAYEGNWNPEAHPPLIHHEHHEEEETPHSTSTIQATPSSTTEETATQKEQWFGNRWHEGYRQTPKEDSHSTTGTAA ASAHTSHPMQGRTTPSPEDSSWTDFFNPISHPMGRGHQAGRRMDMDSSHSITLQPTANPNTGLVEDLDRTGPLSMTTQQSNSQ SFSTSHEGLEEDKDHPTTSTLTSSNRNDVTGGRRDPNHSEGSTTLLEGYTSHYPHTKESRTFIPVTSAKTGSFGVTAVTVGDSNSNV NRSLSGDQDTFHPSGGSHTTHGSESDGHSHGSQEGGANTTSGPIRTPQIPEWLIILASLLALALILAVCIAVNSRRRCGQKKKLVIN SGNGAVEDRKPSGLNGEASKSQEMVHLVNKESSETPDQFMTADETRNLQNVDMKIGV

## Data Availability

Data is contained within the article and Appendix A.

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
