# Peer review of "New Bioinformatic Insight into CD44: Classification of Human Variants and Structural Analysis of CD44 Targeting"

_ijms, 2025, doi:10.3390/ijms26209886_

Round 1
Reviewer 1 Report
Comments and Suggestions for Authors
Gerlicz and co-authors presented a paper on the bioinformatic analysis of variants and structures of CD44, an important receptor that binds hyaluronic acid (HA) and osteopontin (OPN). To my best knowledge, this is the first article in years that systematically describes CD44 variants. This would be of high interest to scientists working in the field. The bioinformatic analysis of CD44 variants was further supported by analysis of available x-ray structures of CD44 in complex with small-molecule fragments. This is also an important part of the paper, since CD44, due to its flat surface is considered as an undruggable protein.
The manuscript is well written. Unfortunately, due to the limited content of experimental methods, it seems to me that this is more of a review paper than a research article. However, the idea of paper is rather good and might be interesting to the scientific community. A valuable extension of the article would be to analyze hotspots on the surface of CD44 (human and murine variants) using computational methods like the FTMap server (e.g., Kitel et al. 2022) to conclude on the druggability of this protein. After the addition of this part to the study, the paper could be published as an article without any further delay. I will be happy to see the revised version of the paper.
Author Response
In the file.

Reviewer 2 Report
Comments and Suggestions for Authors
This is a well-written and timely manuscript providing a thorough classification of CD44 isoforms and an insightful structural analysis of CD44–ligand interactions. The paper consolidates scattered information into a coherent reference and will be of interest to researchers in cancer biology, bioinformatics, and drug discovery.
I recommend publication after minor revision.
I strongly recommend following suggestions to improve the manuscript:
1) Highlight in the main text that the complete isoform mapping (Table S2) and compound tables (Table S5) are available in the Supplementary, since these are highly useful resources.
2) Provide a brief nomenclature guideline in the main text (even a boxed summary), to help readers quickly understand your classification.
3) For the reported micro-splice variants (exon 5 and V3), briefly discuss whether these are strongly supported by sequencing evidence, ideally referencing genomic or RNA-seq support, or could represent annotation artifacts.
4) Add a short comment on human vs murine structural conservation of binding pockets, since many inhibitors are derived from murine data (Table S5).
Best,
Author Response
In the file.
